# Entangled spin-polarized excitons from singlet fission in a rigid dimer

Ryan D. Dill [1,4], Kori E. Smyser [1,4], Brandon K. Rugg[2], Niels H. Damrauer [1,3] ✉ & Joel D. Eaves [1,3] ✉

Singlet fission, a process that splits a singlet exciton into a biexciton, has promise in quantum information. We report time-resolved electron paramagnetic resonance measurements on a conformationally well-defined acene dimer molecule, TIPS-BP1', designed to exhibit strongly state-selective relaxation to specific magnetic spin sublevels. The resulting optically pumped spin polarization is a nearly pure initial state from the ensemble. The long-lived spin coherences modulate the signal intrinsically, allowing a measurement scheme that substantially removes noise and uncertainty in the magnetic resonance spectra. A nonadiabatic transition theory with a minimal number of spectroscopic parameters allows the quantitative assignment and interpretation of the spectra. In this work, we show that the rigid dimer TIPS-BP1' supports persistent spin coherences at temperatures far higher than those used in conventional superconducting quantum hardware.

Quantum information promises advances in science and computing not seen since the revolutions in classical computing that have unfolded over the last 80 years[1]. But unlike classical computing, where the solid-state transistor has become ubiquitous, we remain in the discovery phase for quantum materials. Quantum logic uses qubits that are built upon fragile non-equilibrium quantum states that irreversibly decay to Boltzmann equilibrium. In strong-field experiments, microwave or radio frequencies manipulate the qubits to perform operations[2]. Because the resonant frequencies are much smaller than the thermal energy at room temperature, without extreme cooling or other means of control, a significant population in the excited state generates thermal uncertainty in the initial state of the wavefunction[3]. This "tyranny of temperature" makes quantum circuits classical for temperatures above a few kelvin[4].

Removing the uncertainty in the initial condition of the wavefunction solves the so-called "state-initialization problem," a requirement for quantum computation that DiVincenzo articulated more than 20 years ago[1]. For example, in color centers, like nitrogen-vacancy centers in diamond, a weak-field optical excitation initializes the system into a non-equilibrium state—a magnetic sublevel—where strong-field magnetic resonance pulses perform gate operations[5]. But controlling the placement of such defects in crystals is challenging, which makes scaling the number of qubits in these materials a formidable hurdle. Recent molecular analogs to the color centers suggest that a bottom-up approach from synthetic chemistry might ultimately lead to more scalable architectures[6,7]. Like many other quantum materials, however, those molecules only exhibit quantum function near liquid helium temperatures.

In an earlier theoretical publication, Smyser and Eaves[4] suggested that the state initialization problem might be solved in singlet fission (SF)—a photophysical process that generates a maximally entangled two-triplet state with singlet multiplicity ${}^1TT$—by directing the relaxation from the ${}^1TT$ state into a specific $M$-sublevel of the quintet state, ${}^5TT_M$. They showed that relaxation from ${}^1TT$ populates a pure quintet sublevels when four conditions are met. First, the chromophores should share common molecular axes. Second, the inter-chromophore exchange interaction, $J$, that splits the spin states, must be large. The final two conditions require that the system is spatially ordered and dilute. In practice, these conditions are very difficult to fulfill simultaneously. When they are not satisfied, both static and dynamic sources of decoherence emerge.

For example, in solid-state SF systems, individual chromophores are aligned through crystallization. Triplet pairs may form at sites

[1]Department of Chemistry, University of Colorado Boulder, Boulder, CO 80309, USA. [2]National Renewable Energy Laboratory, 15013 Denver West Parkway, Golden, CO 80401, USA. [3]Renewable and Sustainable Energy Institute, University of Colorado Boulder, Boulder, CO 80309, USA. [4]These authors contributed equally: Ryan D. Dill, Kori E. Smyser. ✉e-mail: niels.damrauer@colorado.edu; joel.eaves@colorado.edu

where $J$ is large, but in crystals the excitons are mobile and hopping unpairs the biexciton state, which can then decohere[8–12]. There have been no reports of an aligned system of molecular SF dimers, where chromophores are covalently coupled through synthetic design. In fact, most molecular dimers are conformationally flexible[13–16], which allows the relative geometry of the two chromophores to fluctuate. Geometrical fluctuations can play a similar role to hopping. If the triplets are independent, there is no quantum advantage to preparing them using SF—they might as well have been prepared through conventional intersystem crossing.

In this article, we take a bottom-up approach and initialize specific quantum spin states of biexcitons at high temperatures using SF in rigid molecular dimers called TIPS-BP1'. We report on the time-resolved electron paramagnetic resonance (trEPR) spectrum from a dilute glass of these dimers. Electron spin coherence between the $^5TT_M$ sublevels produces observable Rabi oscillations in the trEPR spectrum at 75 K. These coherences, which form the basis of quantum gates, persist for microseconds, orders of magnitude longer than the gate switching time for electrons. Because the Rabi oscillations self-modulate the trEPR signal, they allow for a new frequency-selective detection scheme for the spectrum that we report here. Unlike the spectrum from most SF systems in the literature, triplet ($S = 1$) photoproducts are undetectable—the observed spectrum is entirely from the quintet state. Because of the fixed and relatively small inter-chromophore separation in TIPS-BP1' dimers, and because dilution inhibits triplet diffusion, we assume that $J$ is large, that excitons are immobile, and that rare fluctuations in $J$ drive transitions between specific spin sublevels. Under these assumptions, we extend the model from Smyser and Eaves to interpret the solution-phase spectrum of nonparallel chromophores, using only three adjustable spectroscopic parameters. Our results show that strong state selectivity, when the $^1TT$ state transfers into specific $^5TT_M$ sublevels with high fidelity, results if chromophores in a rigid dimer share a single molecular axis. By combining continuous wave (CW) methods and theory, we show how candidate dimers can be tested for their utility in quantum information applications, without pulsed methods and without having to simultaneously solve the immobilization and crystallization problems[17].

## Results

The room temperature transient absorption dynamics of TIPS-BP1' have been previously discussed (see Supplementary Note 1.2)[18]. But they do not distinguish the several possible biexciton species $^{2S+1}TT_M$ that differ in their overall spin $S$, spin projection $M$, and degree of entanglement, so we turn to trEPR to resolve them. The experiment starts the SF process with an optical pulse and then uses EPR to monitor the time-evolution of the products. The trEPR signatures of TIPS-BP1' in mTHF glass (75 K, 640 nm pump wavelength) emerge over a few hundred nanoseconds following photoexcitation (Fig. 1a). This timescale is consistent with the decay of $^1TT$ (Supplementary Figs. 2–4) and is impulsive on the timescale of the trEPR measurement (10 μs)[18], suggesting a small population in the EPR-active state. Four sharp features, from 338–359 mT, dominate the trEPR spectra for all observable times. They form concomitantly and exhibit underdamped Rabi oscillations that beat at the nutation frequency (Fig. 1a, inset). These oscillations have not been reported in trEPR data for any system undergoing SF but have been observed for triplets where relaxation processes are slow[19].

General trEPR trends in the SF literature include broad and congested spectra, with substantial interconversion between EPR-active states[11–13,15–17,20]. By contrast, our spectra—aside from the oscillations and decay—do not show substantial time evolution. They are also highly structured and symmetrical. The EPR spectra in Fig. 1a are narrow, with intensity spanning 20 mT. The intersystem crossing triplet spectrum for the monomer TIPS-Pc (Supplementary Fig. 6), in

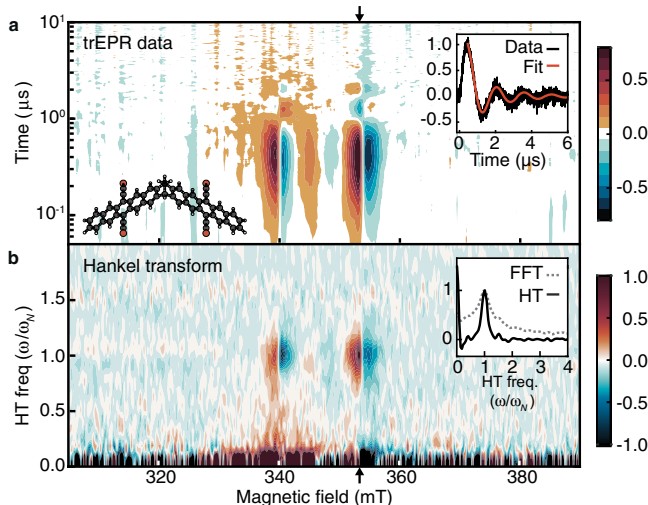

**Fig. 1 | The trEPR data for TIPS-BP1' demonstrates long spin-coherence times.** **a** Contour plot of X-band trEPR data for TIPS-BP1' (75 K and 640 nm excitation). Source data are provided as a Source Data file. The structure of the dimer is shown in the bottom left although the isopropyl groups of the TIPS moieties are removed for clarity (black spheres = C; white spheres = H; red spheres = Si). Inset: Under-damped Rabi oscillations for a representative magnetic field value (353.4 mT, black arrows). The signal decay (black line) fits well to a damped Bessel function (red line), which is expected for an orientationally distributed sample. The fit gives an estimate for the coherence time, $T_2 \approx 1.4$ μs. **b** The Hankel Transform (HT) provides the nutation spectrum at each field point. Inset: The HT of the transient shown in the inset of (**a**) peaks much more sharply than the comparable amplitude spectrum from the Fast Fourier Transform (FFT). This resolution enhancement facilitates extraction of the "Hankel spectrum" in Fig. 2b, which corresponds to $^5TT_0 \leftrightarrow {}^5TT_{\pm 1}$ transitions.

comparison, spans 84 mT. The relatively narrow width of the TIPS-BP1' spectra suggests that the signal originates from $^5TT$[21].

Nutation frequencies depend on $S$ and $M$, so they can, in principle, inform on the spin species and sublevels produced after SF[22]. In the SF literature, they are commonly determined with pulsed EPR at only a few values of the static magnetic field, $B_0$[11–13,15–17]. Compared to pulsed nutation experiments, trEPR has a dramatic multiplex advantage—an entire time trace is collected simultaneously (Fig. 1)[23]. However, pulsed techniques with high microwave powers are necessary for most SF systems since rapid dephasing and population transfer overdamp the low-frequency nutation oscillations in trEPR[24]. In TIPS-BP1', by contrast, the presence of Rabi oscillations at a dominant frequency in the trEPR data implies that there is a state-selective population formed rapidly on the timescale of the oscillation period. Because they appear in the trEPR data, we can directly assign the transitions in it—a significant advantage since that would otherwise require pulsed experiments.

Although Rabi oscillations within a two-level quantum system decay as a damped sinusoid, the measured trEPR nutation signal for an inhomogeneously broadened system does not—it decays as a damped Bessel function[19,22–25]. In such systems, including disordered samples like ours, the measured signal contains contributions from many spins with a distribution of resonant frequencies. Integrating over the distribution of resonant frequencies leads to the result $s(t) \propto J_0(\omega_N t)e^{-t/2T_2}$[25,26], valid for strongly underdamped Rabi oscillations that begin suddenly. Here, $s(t)$ is the time-domain signal for a fixed value of $B_0$, $J_0$ is a zeroth-order Bessel function, $\omega_N$ is the observed nutation frequency, and $T_2$ is the transverse relaxation time, or coherence time. This result requires long $T_2$, a large $B_1$ field strength, or both—note these conditions are different than they are for measurements of the $T_1$ time, and they are satisfied in our system. More information on the measurement appears in Supplementary Note 2.2.

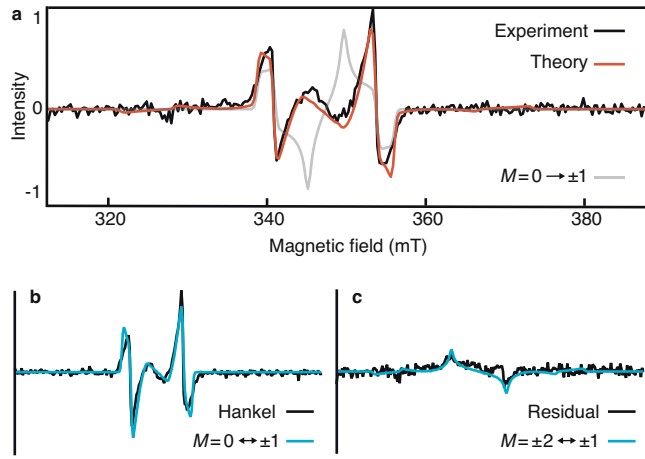

**Fig. 2 | Data and calculated EPR spectrum for TIPS-BP1'. a** The prompt trEPR spectrum for TIPS-BP1' (black) is an average over 200–400 ns (Fig. 1a). The red line comes from the *JDE* model with best-fit parameters $D = 1322 \pm 3$ MHz, $X = 59 \pm 1$ MHz, and $\beta = 111.1 \pm 0.2°$ (see Supplementary Note 2.6). Using the $Q_0$ model to predict initial populations (gray) does not reproduce the spectrum. **b** The Hankel spectrum (black) is the dominant signal and is replicated well by a calculated $^5TT_0 \leftrightarrow {}^5TT_{\pm1}$ spectrum (blue, *JDE* model). **c** The residual spectrum (black) is the difference between the full spectrum in **a** and the Hankel component in **b**. It is reproduced (blue, *JDE* model) with a calculated $^5TT_{\pm1} \leftrightarrow {}^5TT_{\pm2}$ spectrum. Any signal from triplets is undetectable. Relative amplitudes of the calculated spectra in **b** and **c** (blue) are computed from the *JDE* model and are not adjustable parameters.

Because $s(t)$ follows the Bessel function, the Hankel transform, which projects the time-domain signal onto the Bessel functions, substantially enhances frequency resolution relative to the Fourier transform (Fig. 1b, inset, and Supplementary Fig. 8). While Bessel functions approach sinusoids at long times, making use of that fact to use the Fourier transform requires eliminating the strong signal at early times, at a great cost to the signal-to-noise ratio.

Our method shares many similarities with lock-in detection, but rather than externally modulating the signal, the method "locks in" at the dominant nutation frequency $\omega_N$ to separate low-frequency components from the oscillating signal (Fig. 1b). The "Hankel spectrum" is the integrated intensity along the frequency axis within a prescribed bandwidth (see Supplementary Note 2.2). It isolates the signal that nutates at $\omega_N$—the majority component of the EPR data (Fig. 2b, black line).

Fitting the nutation signal at fixed $B_0$ to the expected function $J_0(\omega_N t) e^{-t/2T_2}$ provides a simple way to extract the decoherence time, $T_2 \approx 1.4$ μs (Fig. 1a, inset, and Supplementary Fig. 7). While our application of the Hankel transform to produce a spectral representation of the data is unique, the theoretical and experimental basis for using the Bessel function relationship to measure $T_2$ using trEPR experiments is well established[25,27–31]. While the oscillatory nature of the signal is an unambiguous demonstration of magnetic sublevel coherence, one potential issue is that at intermediate microwave field values, $T_1$ decay processes might impact the signal[25,31]. We verified that increasing the microwave power increases the nutation frequency, as expected, but it does not change the decay rate (see Supplementary Note 2.2), implying that $T_1$ processes do not contribute to the observed decay. The Hankel transform analysis presented here is in many ways complementary to more conventional pulsed echo measurements, also used to measure $T_2$ in inhomogeneously broadened systems. Future work will compare the relative merits between the two types of experiments. Notably, there are two recent works in the literature that use singlet fission in the pursuit of quantum information applications, but with chromophores in crystals that are oriented in the field, not dimers[8,32]. The value of $T_2$ for TIPS-BP1' dimers reported here at 75 K is

similar to that reported in ref. [32] for a single-crystal tetracene-derivative at 10 K, although the methods used to measure $T_2$ are different (ref. [32] used the echo approach). At similar temperatures, the $T_2$ we determine for TIPS-BP1' in a glassy phase, is several times longer than it is in the crystalline samples of ref. [32].

A signal oscillating at a dominant nutation frequency might result from a state-selective relaxation process, from $^1TT$ into a few specific $^5TT_M$ sublevels, and such precise state selectivity can solve the state-initialization problem in quantum information. But to determine the extent of state selectivity in a molecule, an accurate interpretation of the EPR spectrum is essential. Some have adapted Merrifield's theory[33] for triplet-triplet annihilation, to compute the TT populations that the EPR experiment probes (see Supplementary Note 2.3)[10,12,16,32]. Therein, when the inter-chromophore exchange interaction $J$ is zero, the resulting spectrum only comes from $M = 0 \rightarrow M = \pm 1$ transitions, so we refer to it as the '$Q_0$' model. But the $Q_0$ model is inappropriate for strongly coupled dimers, where $J$ is not small. Because $J$ is large in our dimers, the $Q_0$ model does not reproduce the spectrum of TIPS-BP1' (Fig. 2a). Without a theory to determine the populations, they become fitting parameters[11]. In the dense and broad spectra typical of EPR data for SF, additional parameters lead to uncertainty and overfitting that complicates the interpretation of the spectra.[14,34,35]

To overcome this problem, we compute the populations of the initial $^5TT$ sublevels with our nonadiabatic transition theory by extending the theory reported in ref. [4] to model non-parallel chromophores and to compute spectra for dimers in the glass phase (Eq. (1), Methods). Like ref. [4], we assume that $J$, the Dirac-Heisenberg coupling between triplets on adjacent chromophores $A$ and $B$ of the same molecule, is the largest energy scale of the matter hamiltonian and choose the quantization axis to lie along the Zeeman field in the lab frame. This choice diagonalizes the Zeeman hamiltonian and the rotationally invariant, or isotropic, part of the *JDE* hamiltonian for all orientations of the molecule, $g\mu_B B_0 S_z + J\mathbf{S}_A \cdot \mathbf{S}_B$, in the basis of total spin Zeeman states $|S, M\rangle$. The remaining "zero-field hamiltonian" is anisotropic—it depends on a molecule's orientation relative to the quantization axis. Because of the rigidity of the dimer and the relative chromophore orientation, the zero-field hamiltonian becomes a function of three parameters: the axial intra-chromophore interaction, $D$, the anisotropic inter-chromophore interaction, $X$, and the angle between the chromophores, $\beta$ (Supplementary Notes 2.4–2.6, Fig. 3b). A projection operator, $P$, partitions the hamiltonian into the reference hamiltonian, $H_0 \equiv PHP$, that is block-diagonal in total spin $S$, and the perturbation that involves only the anisotropic zero-field hamiltonian $V = H - H_0$.

After this partitioning, the Zeeman term lies on the diagonal of $H_0$ and splits states of different $M$. The exchange interaction $J$ produces diagonal terms that split states with variations in total spin $S$. We assume that nuclear motions modulate the distance between chromophores to make the value of $J$ time dependent, $J = \langle J \rangle + \delta J(t)$.[4] Fluctuations in $J$ modulate the energy gaps between diagonal states of different $S$, bringing them into transient resonances that allow transitions between them. After including the fluctuations on the diagonal by introducing a linear response system-bath hamiltonian and applying perturbation theory in $V$ for the transition rates between states $|\mu\rangle$ and $|\nu\rangle$ of $H_0$, the expression for the rate is analogous to the nonadiabatic Marcus theory, $k_{\mu \rightarrow \nu} \propto |\langle \mu | V | \nu \rangle|^2 F_{\mu,\nu}$, where $F_{\mu,\nu}$ is a temperature-dependent nuclear factor that depends on the statistics of the fluctuations[36,37]. But, because the energy levels of $H_0$ are split by energies much smaller than $k_B T$, the state populations are approximately independent of the statistics for the fluctuations in $J$ and thereby independent of $F_{\mu,\nu}$. In this theory and under the stated assumptions, the tunneling matrix element $|\langle ^1TT | V | \nu \rangle|^2$ gives the sublevel population of the initial state in the trEPR experiment $|\nu\rangle$ from $|^1TT\rangle$.

In the chosen representation, $H_0$ is block-diagonal in $S$, but there is weak mixing between states of different $M$ within an $S$ block. One can

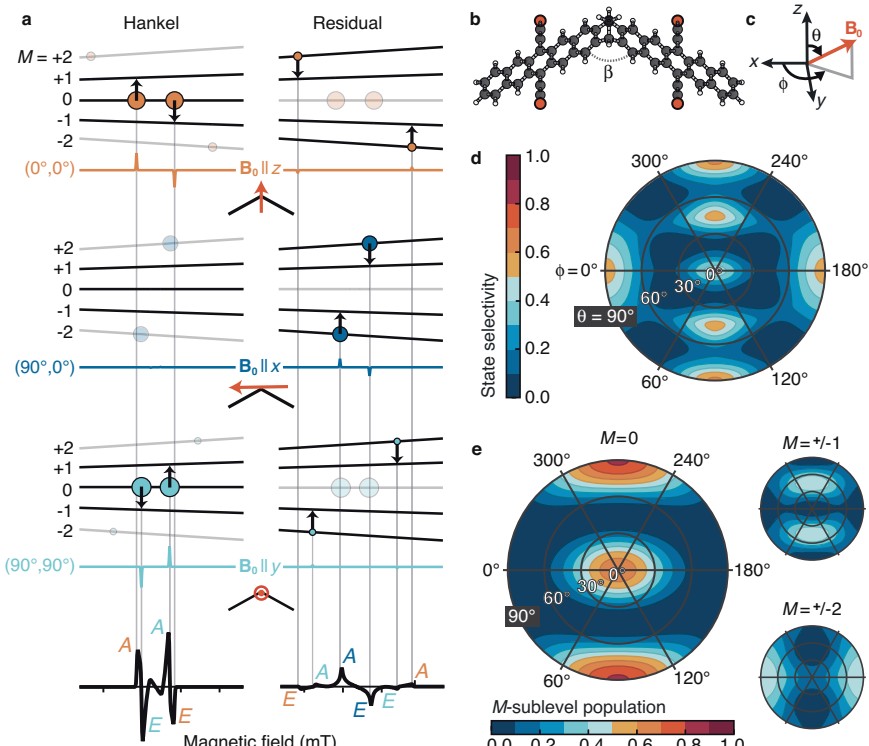

**Fig. 3 | Theory predicts strong state selectivity for the rigid TIPS-BP1' dimer.**
**a** Fixed-orientation EPR spectra (colored lines) for $\mathbf{B}_0$ (red arrows) applied along cardinal dimer directions. Predictions for the Hankel spectrum ($^5TT_0 \leftrightarrow {}^5TT_{\pm1}$) on left and residual spectrum ($^5TT_{\pm1} \leftrightarrow {}^5TT_{\pm2}$) on right. Area of colored circles indicates $^5TT_M$-sublevel populations, which are shown translucent if they do not appear in the corresponding spectrum. Arrows show the direction of transitions ($A$ absorption or $E$ emission). Vertical lines correlate population assignments with features in the simulated powder spectra (black, below). **b** The two chromophores in the TIPS-BP1'
dimer are rigidly linked so that a single bridging angle $\beta$ defines the chromophore-chromophore orientation. **c** The polar and azimuthal angles $\theta$ and $\phi$ for the Zeeman field relative to the dimer cardinal axes. **d** State selectivity order parameter $\mathcal{I} = 1 + \frac{1}{\log_2(5)}\sum_{M=-2}^{+2} p_M \log_2 p_M$ for an ordered sample as a function of the dimer-field orientation. This range of $(\theta, \phi)$ considers all orientations with unique spectra. **e** $^5TT_M$ populations as a function of the dimer-field orientation. The $\pm M$-sublevels are predicted to be equally populated. The maximum population of $^5TT_0$ occurs at $\mathbf{B}_0\|y$, $(\theta, \phi) = (90°, 90°$ or $270°)$.

either ignore the mixing or diagonalize the block, redefining both the states and the transition matrix elements. These two choices correspond to a diabatic basis or an adiabatic one for $|\nu\rangle$, respectively. Unlike the adiabatic states, the diabatic (Zeeman) states are well-defined in the lab frame and independent of orientation (see Supplementary Note 2.6). The diabatic states facilitate assignment of the trEPR spectrum (Fig. 2b, c) but the adiabatic states (Fig. 2a) give a more accurate reproduction of it. Because the applied Zeeman field is much larger than the zero-field interactions, the mixing between the $^5TT_M$ sublevels is weak, and there are only small, quantitative differences between the spectra calculated with the diabatic and adiabatic bases (Supplementary Fig. 12).

To fit spectra, we compute the ensemble-averaged trEPR spectrum directly in the adiabatic basis, use discretization to calculate and sample over molecular orientations, and use simulated annealing to determine best fit parameters. For each orientation in the fitting protocol, the initial states of the trEPR spectra come from the non-adiabatic transition theory. These simplifications are substantial. Namely, the computed trEPR spectra only depend on the parameters $D$, $X$ and $\beta$. Including $X$ is essential but the orthorhombicity parameter $E$ is not included because it does not significantly improve fit results for either the dimer or the monomer (Supplementary Note 2.7). Supplementary Notes 2.4–2.7 contain the details of the theory and the fitting procedure.

Figure 2a shows the prompt EPR spectrum—the spectrum immediately following SF—for TIPS-BP1' (200–400 ns, Fig. 1a and Supplementary Fig. 5) along with a calculation of it. The best-fit values $D = 1322 \pm 3$ MHz and $X = 59 \pm 1$ MHz are consistent with those for

pentacene derivatives and dimers, respectively[17]. Remarkably, the fit value of $\beta = 111.1 \pm 0.2°$ is within 0.2% of the calculated value from DFT simulations for the quintet in a model of TIPS-BP1' (110.9°, unrestricted-$\omega$-B97XD/6-31G(d)).

With the optimal set of spectroscopic parameters determined, the calculated EPR spectrum breaks down into two components from the diabatic $^5TT_M \leftrightarrow {}^5TT_{M\pm1}$ transitions. Figures 2b and c show the results. Our theory demonstrates that the Hankel transform isolates the signal from the $^5TT_0 \leftrightarrow {}^5TT_{\pm1}$ transitions (Fig. 2b), and supports the assignment of the nutation frequency to this component. The residual spectrum (Fig. 2c, Supplementary Figs. 10 and 11), the difference between the Hankel spectrum (Fig. 2b) and the total spectrum (Fig. 2a), agrees with the computed $^5TT_{\pm1} \leftrightarrow {}^5TT_{\pm2}$ spectral component quantitatively, in both amplitude and functional form.

It is only by accounting for the orientational dependence of the sublevel populations that we recover the spectrum from TIPS-BP1'. Figure 3a shows that the most intense features in the powder spectra are from transitions where the Zeeman field aligns with the dimer axes (Fig. 3b, c). Figure 3a also shows that while the $^5TT_0$ sublevel population is large for $\mathbf{B}_0\|z$ and $\mathbf{B}_0\|y$, it is zero for $\mathbf{B}_0\|x$. In the $Q_0$ model, by contrast, the $^5TT_0$ sublevel is the only TT sublevel populated for any orientation—including $\mathbf{B}_0\|x$—leading to an over-representation of the $^5TT_0 \rightarrow {}^5TT_{\pm1}$ transitions in the spectrum, and a poor resulting fit (Fig. 2a, gray). Indeed, if the $^5TT_0$ level were the only sublevel populated, the residual spectrum would be zero.

To engineer a piece of quantum hardware based on our system and observations, one would have to immobilize and align the molecules so that they all have a definite orientation with respect to the

Zeeman field. Figures 3d and e show the predicted state selectivity for a system of aligned TIPS-BP1' dimers. Borrowing an idea from Shannon's classical information theory[38], we introduce the order parameter $\mathcal{I}$ to quantify the state selectivity achievable into all $^5TT_M$ sublevels from $^1TT$ as a function of molecular orientation relative to the field (Fig. 3c), where $\mathcal{I} = 1 + \frac{1}{\log_2(5)} \sum_{M=-2}^{+2} p_M \log_2 p_M$ (Fig. 3d). Like Shannon's information measure, $\mathcal{I}$ is zero when all $^5TT_M$ are equally populated and unity when only one level is occupied. Our work in ref. [4] recommends that the chromophores share a common set of axes. While the $x$ and $z$-axes of the chromophores are not parallel for TIPS-BP1', the $y$-axes are. As a result, the most intense state selectivity occurs when the Zeeman field aligns with the shared $y$-axis[39]. The corresponding north and south poles of Fig. 3d exhibit the largest state selectivity, and Fig. 3e shows that the $^5TT_0$ sublevel is the one that gets polarized.

## Discussion

In molecular systems like those pioneered in nuclear spin resonance computing, scaling the number of coherent qubits is relatively straightforward[2]. But the state-initialization problem has bedeviled that field[3]. TIPS-BP1' is an example of a novel class of compounds that create entanglements between electron spin states that remain coherent on timescales that are orders of magnitude longer (≈1 μs) than expected switching times for a gate operation (typically a few ns), even in a powder spectrum. The quintet state, born under the selection rules of singlet fission, is a two-triplet spin-coherent excitation. The coherence entangles the triplets and increases the number of computational states from three to five—an elementary demonstration of scaling.

Our results highlight several inherent and interrelated design principles worth summarizing. First and foremost, the dimer rigidity statically aligns key magnetic axes, enabling the exploitation of selection rules that depend on inter-chromophore symmetries, thereby dictating which magnetic sublevels of the quintet are produced from the $^1TT$ at any given orientation of the dimer in the Zeeman field. Second, a large mean $J$ ensures that $S$ is a good quantum number—total spin states are energetically well-separated and weakly coupled for all orientations. These features limit quintet population decay by inhibiting decoherence and unpairing (such as decay to two $T_1$, observed in many other crystals and in flexible dimers but not observed here) and by inhibiting mixing with $^1TT$.

Perhaps counterintuitively, the relatively large rate constant for the decay of $^1TT$ to the ground state makes quintet state formation impulsive on the EPR time scale and enhances the Rabi oscillations. In the quantum gates that would follow the optical preparation step, impulsive generation of the quintet would also minimize the phase error in the initialized wavefunction. In a simple kinetic model for the populations of the $^1TT$ and $^5TT$ states, the rise time of the quintet subpopulation is Φ/$k$, where Φ is the quintet quantum yield and $k$ is the intersystem crossing rate from $^1TT$ to a spin sublevel of the quintet. In our system, the quantum yield is low because the decay from $^1TT$ to the ground state is fast (of order 150 ns; Supplementary Note 1.2). While the rise time can also be shortened by increasing $k$ using molecular design, detailed balance for $J/k_BT \ll 1$ implies that the backward rate must be comparable to $k$. Increasing the quantum yield by increasing the intersystem crossing rate would therefore also increase the decay rate from the quintet back to $^1TT$, which is undesirable because it both pollutes state purity and shortens the sublevel coherence time. Thus, impulsive quintet generation comes at the expense of a low quantum yield.

The downside is that low quantum yields may limit versatility, but it remains unclear what the implications are for EPR-based quantum devices whose states are initialized in low yields but with high purity. Synthetic modifications of the molecule might enhance quantum yields by tuning relative rate constants, but care must be exercised to

avoid the coherence-loss pitfalls outlined above. Of course, a quantum device using molecules like TIPS-BP1' would have to immobilize and orient the molecules in a medium so that the Zeeman field would point along the directions of high state selectivity (Fig. 3). Still, our work provides proof of principle for executing such molecular design steps and evaluating their impact on quantum function without also having to simultaneously solve this difficult problem. The results for TIPS-BP1' should motivate more general efforts to employ singlet fission in rigid dimers for quantum information applications.

## Methods

### Transient absorption spectroscopy
Transient absorption data (Supplementary Note 1.2) were collected on a commercially available spectrometer (Ultrafast Systems, EOS) equipped with a continuous flow cryostat (Janis, STVP-100). The sample was prepared under a nitrogen atmosphere in a glovebox and flame sealed under vacuum in an ampoule made from a glass test tube (47729-572). Excitation was at 640 nm (pump fluence < 1 mJ/cm²).

### trEPR spectroscopy
The trEPR data were collected at X-band (9.73 GHz) with a Bruker ELEXSYS-E580 in transient mode (CW microwave source) equipped with a dielectric resonator (Bruker EN 4118X-MD4, Q ≈ 3500) and a closed cycle helium cryostat for low temperature operation. Optical excitation (640 nm, 10 Hz repetition rate, pulse energy ≈ 3.5 mJ/pulse, FWHM pulse duration ≈ 5 ns) was provided by an optical parametric oscillator (Opotek Radiant SE 355 LD) fiber coupled to a reflective collimator (Thorlabs RC08SMA-P01). The collimated beam (diameter ≈ 8.5 mm) was directed through the optical window of the cryostat at the resonator's optical window. The sample was prepared with concentration ≈ 74 μM (A(634 nm) = 0.64 in a 2 mm path length cuvette), transferred to a homemade 4 mm outer diameter quartz EPR tube (1 mm wall thickness), degassed by several freeze-pump-thaw cycles, and then flame-sealed under vacuum with an oxyhydrogen torch. The trEPR data were collected the day following sample preparation.

The default Bruker software (Xepr) permits multi-shot averaging at single field points before stepping the field, but does not permit repeating the entire field-sweep process. We have found that this procedure does not sufficiently average out low-frequency oscillatory background signals. To mitigate this problem, we used a custom python script (run through the Xepr API) provided to us by Bruker (Ralph Weber), that allows repeating the entire field-sweep. The large time separation between repeated measurements at each field point substantially reduces the oscillatory background signals, which seem to maintain a stable phase relationship with the laser pulse train only over short durations. This procedure also facilitates detection of sample degradation during the experiment, but can produce very large data sets, and increases measurement time due to the non-zero field stepping time. A total of 6 two-dimensional scans (time × field) were collected and averaged together.

### Data analysis
TA data and trEPR data were both processed with MATLAB code written in-house. Spectral fitting and simulations were performed with code written in Julia.

The trEPR data were background corrected and then rephased to extract the absorptive signal. For Hankel transform analysis, the processed absorptive signal was cropped at the first signal maximum, $t_0 = 401$ ns, and multiplied by a one-sided Hann window function of 10 μs duration which was zero-padded to a total duration of 65,536 ns (the original recorded signal duration). We then applied the Hankel transform to each transient (i.e. kinetic trace) in the windowed two-dimensional data set (time × field) to generate the two-dimensional

spectrum (HT frequency × field) pictured in Fig. 1b. The Hankel component spectrum in Fig. 2b is a weighted average of the HT spectra in the HT frequency range surrounding the dominant nutation band, with weightings defined by a Lorentzian fit to the HT spectrum in the Fig. 1b inset, further modified by completely excluding data below 2 mrad/ns and above 6 mrad/ns. For more details, refer to Supplementary Note 2.2.

## Simulations

The starting point for the simulations used to interpret and assign the trEPR spectra is the *JDE* hamiltonian for acene-based dimers,

$$\mathcal{H} = g\mu_B B_0 (S_{Az} + S_{Bz}) + J\mathbf{S}_A \cdot \mathbf{S}_B + \mathbf{S}_A^\mathsf{T} \cdot \mathbf{X} \cdot \mathbf{S}_B + \mathbf{S}_A^\mathsf{T} \cdot \mathbf{D}_A \cdot \mathbf{S}_A + \mathbf{S}_B^\mathsf{T} \cdot \mathbf{D}_B \cdot \mathbf{S}_B. \tag{1}$$

The first term in Eq. (1) is the usual Zeeman term where $g\mu_B$ converts the spin to a magnetic dipole moment and $B_0$ is the Zeeman field strength[40]. The single-chromophore spin operators corresponding to chromophore $A$ and $B$ are $\mathbf{S}_A$ and $\mathbf{S}_B$, respectively, and $J$ is the inter-chromophore Dirac-Heisenberg isotropic exchange interaction between them. This term splits states that have different total spin $S$ and does not depend on orientation. Provided that the mean of $J$ is large compared to both $D$ and $X$, the spectrum is insensitive to both the sign and magnitude of $J$. To replicate both the lineshape and intensities in the trEPR spectrum, an anisotropic inter-chromophore interaction $\mathbf{X}$ must be included. $\mathbf{D}_A$ and $\mathbf{D}_B$ are zero-field splitting tensors that correspond to the intra-chromophore anisotropic (spin-dipole) inter-actions and depend on the zero-field scalar parameter $D$. All anisotropic tensors, $\mathbf{D}_A, \mathbf{D}_B$, and $\mathbf{X}$, depend explicitly on the dimer's orientation relative to the Zeeman field (Fig. 3c), while $\mathbf{D}_A$ and $\mathbf{D}_B$ also explicitly depend on the bridging angle $\beta$ that defines the orientation of the two chromophores (Fig. 3b) relative to one another. For this system, the zero field parameters $E$ and $X$ are both small and have little bearing on the initial populations[4]. A nonzero $E$ does not change the spectrum and leads to overfitting (Supplemental Information 2.7) but $X$ has a dramatic impact on it.

For a given molecular orientation, we calculate the hamiltonian in the adiabatic or diabatic basis, employ the short time approximation discussed above to compute the initial populations, and sum over states to compute the intensities. The spectrum results from summing all such orientationally dependent spectra with orientations drawn over the surface of a sphere (Supplemental Information 2.4–2.6) and is a function of the parameters $\beta$, $D$ and $X$. We find the parameters of best fit by minimizing the difference between the short-time data (Fig. 2a) and the calculated spectrum in the weighted least-squares sense, $\chi^2(\beta, D, X) = 1/N \sum_i^N \left[ (S_{i,\mathrm{data}} - S_{i,\mathrm{model}})/\sigma_i \right]^2$, where $S_{\mathrm{model}}$ is the calculated spectrum at the same $N$ field points as the short-time data, $S_{\mathrm{data}}$. The error in the intensity at each field point, $\sigma_i$, comes from jackknifing the average of six consecutively acquired data sets for TIPS-BP1'. We used the SAMIN simulated annealing algorithm from the open source Julia package Optim.jl[41] to find the global minimum of the objective function, $\chi^2$.

## Data availability

Source data for Fig. 1a have been deposited in the figshare platform and can be found at https://doi.org/10.6084/m9.figshare.21781118. They are also provided with this paper as a Source Data file. The experimental data in Fig. 2 is included with the program at https://github.com/joeleaves/JDE.jl. All computed data from Figs. 2 and 3 are generated by the provided program using the specified values for spectroscopic parameters. Source data are provided with this paper.

## Code availability

The source code used in preparation of this article may be obtained from https://github.com/joeleaves/JDE.jl.

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

## Acknowledgements

We thank Dr. Justin Johnson and Dr. Obadiah Reid for discussions. The dimer TIPS-BP1' was synthesized previously by Ethan Miller in a collaboration of author N.H.D. with Professor Tarek Sammakia of CU Boulder and funded by the National Science Foundation (CHE-166537; N.H.D.). We acknowledge funding from the United States Department of Energy, Office of Basic Energy Sciences (ERW7404; N.H.D. and J.D.E.) and from the National Science Foundation (CHE-2102713; N.H.D.). Time-resolved EPR work was supported by the U.S. Department of Energy, Office of Science, Office of Basic Energy Sciences, Division of Chemical Sciences, Geosciences, and Biosciences, under Contract No. DE-AC36-08GO28308. This work also utilized resources from the University of Colorado Boulder Research Computing Group, which is supported by the National Science Foundation (awards ACI-1532235 and ACI-1532236), the University of Colorado Boulder, and Colorado State University. This work was authored in part by Alliance for Sustainable Energy, LLC, the manager and operator of the National Renewable Energy Laboratory. The views expressed in the article do not necessarily represent the views of the DOE or the US government. The US government retains and the publisher, by accepting the article for publication, acknowledges that the US government retains a nonexclusive, paid-up, irrevocable, worldwide license to publish or reproduce the published form of this work, or allow others to do so, for US government purposes.

## Author contributions

R.D.D. and B.K.R. performed the measurements and K.E.S. implemented the theory of K.E.S and J.D.E. Authors N.H.D and J.D.E. advised on all efforts. R.D.D., K.E.S., N.H.D., and J.D.E. contributed to the data analysis and manuscript preparation.

## Competing interests

The authors declare no competing interests.
