## [Peer Review File · Nature Communications]

Entangled Spin-polarized Excitons from Singlet Fission in a Rigid DimerEditorial Note: This manuscript has been previously reviewed at another journal that is not operating a transparent peer review scheme. This document only contains reviewer comments and rebuttal letters for versions considered at *Nature Communications*. Mentions of prior referee reports have been redacted.

REVIEWER COMMENTS

Reviewer #1 (Remarks to the Author):

The authors have addressed all my previous comments adequately. I feel that the modifications and added explanations have increased the value and impact of the work significantly. Consequently, I can recommend this revised version of the manuscript for publication in *Nature Communications*.

Reviewer #2 (Remarks to the Author):

In this paper, the authors report time-resolved electron paramagnetic resonance measurements on a molecule, TIPS-BP1' designed to exhibit strongly state-selective relaxation to specific magnetic spin sublevels showing that the spin polarization is a nearly pure initial Q0 state from the ensemble. The rigid, covalently bound dimer supports persistent spin coherences at temperatures far higher than those used in conventional quantum hardware. This version clearly describes important issues and novel concepts for the applications to the quantum information science using the singlet fission processes of the Rigid Dimer from several experimental results including the transient absorption measurement. Thus, the paper is basically acceptable for *Nat. Commun.*. However, this reviewer feels that the following sentences are not enough clear for the readers on the long-lived quintet when the singlet TT deactivation is quick.

1. At the final part of the revised main text, the authors describe as follows,

"Finally, although perhaps counterintuitive, the sufficiently large rate constant for the direct decay of 1TT to the ground state enhances the quintet sublevel coherences, witnessed as Rabi oscillations, by making quintet state formation impulsive on the EPR time scale. By kinetics, the given by $\Phi/k_{S \rightarrow Q}$, where Φ is the quintet quantum yield. In our system, the decay of 1TT to the ground state is relatively fast (Supplementary Note 1.2). Thus, impulsive quintet generation comes at the expense of a low quantum yield. While the rise time can also be shortened by increasing $k_{S \rightarrow Q}$, detailed balance for $J/k_{BT} \ll 1$ implies that the backward rate $k_{Q \rightarrow S}$ would be comparable, and decay from the quintet back to 1TT is undesirable because it pollutes state purity. In the quantum gate that would follow the optical preparation step reported here, impulsive generation of the quintet would minimize the phase error in the initialized wavefunction."

 The impulsive 5TT_0 generation probably stems from the quick singlet deactivation competing with coherent singlet <-> quintet mixing in the presence of the large exchange coupling. However, this reviewer believes that 5TT_0 state should also decay quickly under this situation. When the singlet <-> quintet relaxation (i.e. equilibrium) is competing with the quick singlet-depopulation process, the quintet decay also needs to be quick even though the quintet state is created "impulsively". The present transient absorption results suggest that the long-lived quintet state do not interact with the singlet manifold, although the quintet generation is caused by the singlet-quintet interaction via the ZFS. I wonder why such isolation of the long-lived 5TT_0 state occurs in the present dimer sample?

By the way, the transient absorption experiment was performed to obtain the direct decay of 1TT to the ground state. But temperature is higher in the TA experiments than the temperature for the TREPR measurement.

Do the authors perform transient nutation experiments at the elevated temperature?
If the quintet EPR decays quick as in the decay of the TA at elevated temperature, the paper is acceptable.

Reviewer #3 (Remarks to the Author):

There is no doubt that the authors have carefully addressed all individual points raised by the four previous referees. The resulting changes and clarifications have certainly improved the quality of the manuscript. However, on a general level, all four referees have expressed their concern that novelty of the results and ideas presented in this manuscript may not be sufficient to warrant publication in [redacted]. Also papers to be published in Nature Communications are supposed to "represent important advances of significance to specialists within each field". My personal view is that also the revised version of the manuscript does not fulfil this condition.

The referees point out the importance of spin-relaxation time constants (in particular T_2) for real-world quantum information applications, and allude to potential problems associated with determining reliable values for T_2 based on transient EPR measurements. I understand that, as the authors argue, T_2 can be extracted from transient nutation experiments under certain conditions. However, I am also aware of the fact that many factors can indeed cause substantial differences between T_2 measured through transient EPR and (pulse) echo-decay measurements. Given the strong emphasis the authors put on quantum-information aspects, I fully agree with Reviewer #1 that pulse EPR measurements would be very useful to confirm the coherence times reported in the manuscript and to demonstrate that quantum-gate operations on the timescale of a few nanoseconds are feasible.

Second Response Letter to Reviewers for NCOMMS-22-33088A

Joel D. Eaves and Niels H. Damrauer

We thank the Reviewers and the Editor for their time on this manuscript. We have taken their suggestions seriously and made some changes to the document to improve clarity and increase specificity for the audience. Quotes from the reviewers are in red, manuscript edits are in green and previous text in the manuscript is in blue.

Response to Reviewer #2.

Reviewer #2 raises two separate points. First,

The impulsive $5TT_0$ generation probably stems from the quick singlet deactivation competing with coherent singlet \leftrightarrow quintet mixing in the presence of the large exchange coupling. However, this reviewer believes that $5TT_0$ state should also decay quickly under this situation. When the singlet \leftrightarrow quintet relaxation (i.e. equilibrium) is competing with the quick singlet-depopulation process, the quintet decay also needs to be quick even though the quintet state is created "impulsively". The present transient absorption results suggest that the long-lived quintet state do not interact with the singlet manifold, although the quintet generation is caused by the singlet-quintet interaction via the ZFS. I wonder why such isolation of the long-lived $5TT_0$ state occurs in the present dimer sample?

As we discussed in our previous rebuttal, in this paper and in two other publications, the mechanism is incoherent, not coherent. We do not need to worry about coherences between the 1TT and 5TT_M states. The Reviewer is also thinking of the rate model in the long-time limit, not the short-time limit, which is incorrect. At early times, the quintet decay does not need to "*also...be quick*" because the system never has time to come to steady-state between the singlet and quintet, so the rate of the backwards process is irrelevant at early times. It is an elementary calculation to analyze the kinetic model we describe in the text, so we do not think including it in the document is necessary. Nonetheless, we view the relatively fast decay to ground state as an important design principle that our system has revealed and have rewritten one of the concluding paragraphs to highlight it.

308 Perhaps counterintuitively, the relatively large rate constant for the decay of 1TT
309 to the ground state makes quintet state formation impulsive on the EPR time scale
310 and enhances the Rabi oscillations. In the quantum gates that would follow the opti-
311 cal preparation step, impulsive generation of the quintet would also minimize the
312 phase error in the initialized wavefunction. In a simple kinetic model for the popu-
313 lations of the 1TT and 5TT states, the rise time of the quintet subpopulation is Φ/k ,
314 where Φ is the quintet quantum yield and k is the intersystem crossing rate from 1TT
315 to a spin sublevel of the quintet. In our system, the quantum yield is low because the

Second Response Letter to Reviewers for NCOMMS-22-33088A

Joel D. Eaves and Niels H. Damrauer

316 decay from ^1TT to the ground state is fast (Supplementary Note 1.2).
317 While the rise time can also be shortened by increasing k using molecular design,
318 detailed balance for $J/k_B T \ll 1$ implies that the backward rate must be comparable
319 to k . Increasing the quantum yield by increasing the intersystem crossing rate would
320 therefore also increase the decay rate from the quintet back to ^1TT , which is unde-
321 sirable because it both pollutes state purity and shortens the sublevel coherence time.

322 Thus, impulsive quintet generation comes at the expense of a low quantum yield.

The transient absorption data indicate that the quantum yield is low—on the order of a few percent. This implies that the intersystem crossing rate from singlet to quintet, k , must be much smaller than the rate for returning to the ground state. The result is a fast rise time at the expense of a low quantum yield. While the intersystem crossing rate might be made faster and further reduce the rise time of the quintet, detailed balance implies that doing so would also increase the reverse rate, pushing the system closer to the limit that the Reviewer imagines, which would be unfavorable. All of this should now be clear in the paper.

Finally, Reviewer #2 says:

By the way, the transient absorption experiment was performed to obtain the direct decay of ^1TT to the ground state. But temperature is higher in the TA experiments than the temperature for the TREPR measurement.

*Do the authors perform transient nutation experiments at the elevated temperature?
If the quintet EPR decays quick as in the decay of the TA at elevated temperature, the paper is acceptable.*

We interpret that ‘elevated temperature’ in this reviewer comment refers to 102K where the TA data shown in Supplementary Figure 2 were collected as opposed to the 75K where the trEPR data were collected. This small temperature difference is due to a limitation in our instrumentation. The full TA spectra cannot be collected down to 77 K on our instrument. But in our other observations using single-wavelength kinetics measurements, the TA decay dynamics that we observe are temperature independent in this lower temperature regime ($\sim 110\text{K}$ to 77K). Thus, we fully anticipate that the dynamics shown in Supplementary Figure 2 (102K; where we were able to collect a full spectrum versus time data set) will be virtually identical to dynamics at 75K. We have made several changes. First, we modified the caption to the supplementary figure as shown below.

Supplementary Fig. 2: Nanosecond TA Spectra of TIPS-BP1' in mTHF at 102 K. ^1TT forms within a few picoseconds². On a nanosecond timescale, the TA dynamics mainly show decay of ^1TT to the ground state (black arrow). Under these conditions, the long-lived states probed in the trEPR experiment are not seen above the noise, suggesting the observed trEPR signals are from small

Second Response Letter to Reviewers for NCOMMS-22-33088A

Joel D. Eaves and Niels H. Damrauer

populations in those states. As described in the text, the spectral evolution can be modeled with a single exponential decay with a time constant of 145 ns. Using single-wavelength measurements at the peak of the 519 nm excited state absorption (ESA), we observe temperature independence in the decay within the temperature regime of $\sim 110\text{K}$ to 77K .

Second and third, we added two figures and their respective captions as shown below. The first figure added (Supplementary Fig. 3) is an Arrhenius-style plot of $\ln(1/\tau)$ versus $1/T$ that shows the onset of temperature independence in the observed ^1TT lifetime below $\sim 110\text{K}$. The second figure added (Supplementary Fig. 4) is an overlay of single wavelength data collected in the same spectral region at 79K versus 102K (extracted from the full spectrum versus time matrix that underlies Supplementary Fig. 2). We have also changed the main text in trivial ways to accommodate the addition of these new figures when we call out Supplementary Figures.

Supplementary Fig. 3: Arrhenius plot of ^1TT decay for TIPS-BP1' in mTHF. Orange data points are from global fits of wavelength- and time-resolved data like that presented in Supplementary Figure 2. Blue data points are from fits of single wavelength data sets collected on a separate TA setup. Error bars are 95% confidence interval computed from the fit Jacobian using MATLAB's `nlparci` function. At low temperatures (below $\approx 130\text{K}$), the spectral evolution can be modeled with a single exponential decay whose time constant is approximately temperature independent within the temperature range of $\approx 110\text{K}$ to 77K , as seen by the plateau at high values of $1/T$ (example kinetics are shown in Supplementary Figure 4)

Supplementary Fig. 4: Transient absorption kinetic traces for TIPS-BP1' in mTHF. The blue trace is a single wavelength kinetic trace selected from the TA data matrix represented spectrally in Supplementary Figure 2 (522.5 nm probe; 102 K). The orange trace is a single wavelength kinetic trace measured with a monochromator, on a different TA setup, at the same probe wavelength. It is clear from these data that the dynamics at 79 K and 102 K are very similar, which is more easily seen in Supplementary Figure 3

We would like to emphasize and restate in a different way a point made above in response to Reviewer 2's first point because it also addresses this temperature comment. We believe that this Reviewer's comment about the temperature stems from a misunderstanding about the impacts of the simple kinetic model. When there is a smaller rate constant for ^1TT to quintet (k) compared to the rate constant for ^1TT to ground state (k_0), the quantum yield of formation of the quintet is impacted as we have stated in the text. However, once the quintet is formed at low yield, its lifetime is tied to the reverse rate constant quintet to ^1TT (call it k'). Provided that k' is substantively smaller than k_0 , which we fully expect in this system based on the low quantum yield and detailed balance, the quintet will not decay with anywhere near the rapidity associated with k_0 .

Second Response Letter to Reviewers for NCOMMS-22-33088A

Joel D. Eaves and Niels H. Damrauer

Response to Reviewer #3.

First, we have tried to eliminate misunderstandings between pulsed echo experiments and the trEPR methods that we present to measure the dephasing time T_2 . Provided that certain conditions hold, which we demonstrate in our system, both methods can both measure the T_2 time in an inhomogeneously broadened system. We note that Reviewer #1, who seems to know pulsed magnetic resonance well, now wholeheartedly recommends publication and no longer believes that the pulsed experiments are necessary to demonstrate this system as a molecular candidate for further exploration in quantum information. Reviewer #3 now, unfortunately, has taken this position, but we would offer the same reasons for disagreeing with them that we have already provided in our previous rebuttal and that convinced Referee #1. In Reviewer #3's response,

...and allude to potential problems associated with determining reliable values for T_2 based on transient EPR measurements. I understand that, as the authors argue, T_2 can be extracted from transient nutation experiments under certain conditions. However, I am also aware of the fact that many factors can indeed cause substantial differences between T_2 measured through transient EPR and (pulse) echo-decay measurements.

The Reviewer is unspecific about the “potential problems” and “many factors” that cause discrepancies between T_2 measured as we have done it and T_2 measured in the echo experiments. We suspect they mean that T_1 processes can contribute to measured relaxation dynamics or that we might be measuring some other unspecified field-dependent relaxation. In the Supplemental Information, we reported that the decay time is insensitive to the magnetic field strength, though the nutation frequency scales with it in the expected way. We thought this was important enough to move into the main text. The paragraph in question now reads

154 Fitting the nutation signal at fixed B_0 to the expected function $J_0(\omega_N t)e^{-t/2T_2}$
155 provides a simple way to extract the decoherence time, $T_2 \approx 1.4 \mu\text{s}$ (Fig. 1a, inset,
156 and Supplementary Fig. 5). While our application of the Hankel transform to produce
157 a spectral representation of the data is unique, the theoretical and experimental basis
158 for using the Bessel function relationship to measure T_2 using trEPR experiments is
159 well established.^{25,27–31} While the oscillatory nature of the signal is an unambiguous
160 demonstration of magnetic sublevel coherence, one potential issue is that at inter-
161 mediate microwave field values, T_1 decay processes might impact the signal.^{25,31}
162 We verified that increasing the microwave power increases the nutation frequency,
163 as expected, but it does not change the decay rate (see Supplementary Note 2.2),
164 implying that T_1 processes do not contribute to the observed decay. The Hankel trans-
165 form analysis presented here is in many ways complementary to more conventional

Second Response Letter to Reviewers for NCOMMS-22-33088A

Joel D. Eaves and Niels H. Damrauer

¹⁶⁶ pulsed echo measurements, also used to measure T_2 in inhomogeneously broadened
¹⁶⁷ systems. Future work will compare the relative merits between the two types of
¹⁶⁸ experiments.

It would be interesting to compare the echo experiments to this one, but it is outside the scope of the paper and is not within our capabilities to do now in any case. We stress again, however, that the use of trPER to extract T_2 is *not new* and have provided several references—including canonical texts on spin resonance—that discuss it in detail. This is the first work that measures trEPR on rigid dimers and we have not done the pulsed experiments, nor has anyone else. The relative merits between the two types of experiments in terms of sensitivity, signal to noise, and so on, are likely quite system-dependent. We hope these unknowns inspire future work in our field. The changes we have made should address any doubts about the claims of our innovations and the uncertainties of our measurements.

Reviewer #2 (Remarks to the Author):

I appreciate the detailed replies from the authors and the manuscript is clearly improved due to the temperature dependence of the transient absorption data. This reviewer fully understands the reason why the isolation of the long-lived $5TT_0$ state occurs in the present dimer sample; the system never has time to come to steady-state between the singlet and quintet. This situation is however fulfilled only when the quintet TT energy is significantly smaller than that of the singlet TT energy; i.e. the case of the positive J coupling. I believe that it is readable for broad audience that the authors add the such explanation that the large positive J coupling will induce the impulsive and long-lived quintet around line 308 in the main text.